# Takotsubo Syndrome: Differences between Peripartum Period and General Population

**DOI:** 10.3390/healthcare12161602

**Published:** 2024-08-12

**Authors:** Stavros Tzerefos, Dimitra Aloizou, Sofia Nikolakopoulou, Stavros Aloizos

**Affiliations:** 1ICU Department, IASO General and Maternity Hospital, 151 23 Athens, Greece; dimitraloi@yahoo.gr (D.A.); saloizos@yahoo.com (S.A.); 2Emergency Department, Asklepieion General Hospital, 166 73 Voula, Greece; sofnikol@yahoo.gr

**Keywords:** reverse takotsubo syndrome, pregnancy, peripartum stress-induced cardiomyopathy

## Abstract

Takotsubo syndrome (TTS) was first described in postmenopausal women with transient regional wall motion abnormalities. The trigger is usually an emotional or physical stress. The catecholamine hypothesis seems to be the most prevailing. The main difference between TTS and acute coronary syndromes is that there is no obstructive coronary disease to explain the regional abnormalities. In this form, the left ventricle resembles the fishing jar which is used to trap octopus in Japan. However, to date more atypical forms are recognized. Also, the syndrome is not limited to older women. Nowadays, TTS is presented even in pregnancy and postpartum females. Our experience revealed cases of patients during these periods and some of them suffered from reverse Takotsubo. Additionally, the initial diagnosis in some patients was other than TTS. Due to these findings, we suggest that this type of TTS is not very rare but underestimated. For this reason, further studies are needed to support and explain this condition.

## 1. Introduction

Takotsubo syndrome was first introduced in Japan when a female presented with acute chest pain, electrocardiographic (ECG) changes but the coronary angiography was negative for coronary artery disease. The left ventricle (LV) had a transient unusual appearance with a narrow neck and apical ballooning in systole [1]. After this publication, TTS gained awareness globally and has been diagnosed more frequently worldwide. Takotsubo syndrome derived its name from the Japanese word for octopus trap (Figure 1), due to the shape of the LV at the end of systole and has been described under a remarkable number of different names in the literature including “broken heart syndrome”, “stress cardiomyopathy”, and “apical ballooning syndrome” [2].

The cardiomyopathy occurs in the setting of a catecholamine surge due to acute stress leading to cardiac dysfunction. The etiology of the stress factor is often physical or emotional. An emotional, physical, or combined trigger can precede the TTS event. In recent years, new variants of TTS have been described.

We have observed that the reverse TTS type is often presented in the peripartum period (including late pregnancy and postpartum). Pregnancy and postpartum periods are conditions where a woman’s body is under stress. The prevalence of TTS in peripartum is unknown and not well investigated. We have an experience of nine cases admitted to the ICU due to TTS. Our aim is to discuss TTS in the peripartum period and if the reverse type is more common in pregnancy and postpartum compared to the general population as well.

## 2. Our Experience

During the last 5 years in our ICU department, which supports a major obstetric hospital, we have seen nine cases of TTS during pregnancy and the postpartum period. The youngest woman was 28 years old and the oldest 40 years old, respectively. The patients’ features are presented in Table 1. All women had no medical history before conception. One woman was obese. During pregnancy, no patient suffered from any health problems. The vast majority presented with shortness of breath and pulmonary edema and two women complained of chest pain. Most of them had leg swelling and fluid retention. Hemodynamically, they did not have signs of cardiogenic shock, low cardiac output, or left ventricular outflow tract obstruction. The assessment for preeclampsia was negative.

All patients underwent successful delivery and were admitted to our ICU for further diagnostic workup and treatment. We should mention that in five cases the initial diagnosis before admission in the ICU was not TTS. The suggested diagnoses were acute coronary syndrome, coronary dissection, myocarditis, and in one case the possible etiology was non-cardiac. The patients in the beginning underwent ECG and full laboratory blood exams including hs-Troponin and ProBNP. The ECGs were abnormal (Figure 2). Most of them had no specific diffuse ST alterations. One woman presented with ST elevation and the other two with ST depression. The abnormal initial ECGs led to a thorough investigation. Both cardiac markers were elevated. Troponin was mild and increased disproportionately to the extent of myocardial damage. In contrast, the ProBNP levels were high reflecting the left ventricle stress.

An urgent echocardiogram was performed after the admission to the ICU. In six cases, the echocardiogram revealed regional wall motion abnormalities in the mid and apical segments of the left ventricle, raising high suspicion of TTS. However, we faced diagnostic concerns for the other three women. After thorough and cautious study, we concluded that these patients suffered from reverse TTS (Figure 3). This rare type presents with hypokinesia/akinesia of basal segments only. Their ejection fraction was impaired but less than the typical TTS. The echocardiographic findings were compatible with TTS according to the proposed InterTAK diagnostic criteria (Table 2). The next step to establish the diagnosis of TTS was to exclude coronary artery disease. Four postpartum women underwent coronary angiography, another two underwent cardiac computed tomography angiography, and the other three denied further investigation. All nine patients had, as suspected, normal coronary arteries (Figure 4). There were no findings of atherosclerosis or suspicion of spontaneous coronary artery dissection which can occur during pregnancy. The woman with suspected myocarditis underwent a cardiac MRI which was negative (Figure 5). Postpartum cardiomyopathy was included as an alternative diagnosis, but in our patients was less likely due to rapid recovery and echo findings. Echocardiograms were performed every day to monitor heart function and to reveal possible complications. Fortunately, the wall motion abnormalities improved rapidly within a few days and the ejection fraction was restored completely.

Due to similar clinical presentation, the patients received common treatment. We administrated diuretics, nitroglycerin, b-blockers, and angiotensin-converting enzyme inhibitors.

After 3–4 days, the patients were transferred to the department. They were asymptomatic, with significant improvement of heart contractility and without complications. These postpartum women recovered completely and were discharged from the hospital with no symptoms or complications.

Finally, 6 months after the diagnosis we contacted seven of nine women for follow-up information. They were all alive without any symptoms or adverse events. Two patients were lost from follow-up. The frequency of reverse TTS seems to be higher in our observation study compared to the general population, because initially this type was misdiagnosed.

## 3. Epidemiology

TTS represents 1–3% of all ST-elevation myocardial infarctions (STEMIs) and in female patients it is increased to 5–6%. Of all patients, 90% were women with a mean age of 67–70 years old. However, TTS is also described in children [4]. Its occurrence in pregnancy is uncommon and exact incidence is unknown. Some studies show that the incidence of peripartum cardiomyopathy (PPCM) is 1 in 1000 to 4000 births which makes it challenging to distinguish it from TTS [5]. A recent study by Mogos FM et al. reports that from 19,754,535 pregnancy-associated hospitalizations in the USA, 590 were TTS-associated (0.003%) [6].

## 4. Pathophysiology

The pathophysiology of TTS is not completely understood. One of the most well-known and accepted hypotheses concerning the etiology of ventricular dysfunction is the catecholamine theory [7]. Usually, there is an identifiable physical or emotional trigger that precipitates TTS. The syndrome can be caused even by intravenous administration of catecholamines and beta-agonists [8]. It has been shown that patients with TTS triggered by emotional stress have markedly elevated levels of catecholamines compared to patients with myocardial infarction [9]. Analyses have also demonstrated a sympathetic predominance and marked depression of parasympathetic activity during the acute phase [10,11]. Amariles and Cifuentes suggested that acute stressors lead to an increase in the concentration of neuropeptides and catecholamines (dopamine, epinephrine, norepinephrine) during the acute phase of TTS, contributing to LV dysfunction. The elevated levels of catecholamines contribute not only to myocardial dysfunction but also to coronary microvascular vasospasm, increasing cardiac workload and leading to a supply-demand mismatch. The acute mismatch is followed by post-ischemic stunning of the myocardium, resulting in the typical apical ballooning of the left ventricle [12].

The effect of catecholamine excess via increased sympathetic stimulation is the dominant theory for the syndrome but the mechanism that is responsible for the variety of regional ballooning patterns is unknown. There are many explanations such as plaque rupture. Unfortunately, this hypothesis of transient ischemia due to rupture followed by rapid lysis cannot explain the phenomenon. Optical coherence tomography failed to identify plaque rupture in the vast majority [13]. Another proposed mechanism is a multi-vessel epicardial spasm mediated by catecholamines. Patients suffering from TTS have reduced endothelium-dependent dilation [14]. Although epicardial constriction is present in some patients, the majority did not react with spasms using provocative agents.

The effect of catecholamines and endothelin take place in the coronary microvasculature via the a_1_-receptors and endothelin receptor type A. This suggests that microcirculatory dysfunction has an essential role in TTS. Microcirculatory dysfunction in the acute phase of TTS is transient and its recovery appears to correlate with improved myocardial function. Endomyocardial biopsies have revealed contraction band necrosis supporting the theory of direct damage of catecholamines on cardiomyocytes. The viability of myocyte is reduced and occurs via complicated pathways [15].

Especially during pregnancy, a woman’s body undergoes stressful physiological changes. At the late stage of pregnancy, high levels of estrogen are released; after the placenta expulsion, these levels are rapidly exhausted. Delivery is by itself an emotional and physical stress condition that increase catecholamine levels. The decreased levels of estrogen cause the myocardium to become more susceptible to catecholamines [16]. After cesarean delivery, the administration of beta-receptor agonists that are used to stimulate uterine contraction may lead to TTS [17,18,19]. Postmenopausal women have a higher prevalence of TTS; however, younger females have a higher incidence of reverse TTS [20].

In summary, novel evidence supports that TTS is due to acute, excessive release of catecholamines or drug administration affecting patients with susceptible microcirculation and myocytes leading to transient dysfunction.

## 5. Symptoms

Usually, patients in the acute phase complain about chest pain indistinguishable from myocardial infarction or symptoms of heart failure such as dyspnea and acute pulmonary edema. A minority of them are asymptomatic only with positive biomarkers and/or ECG abnormalities. Some signs contribute to complications of TTS like cardiogenic shock, left ventricular outflow tract obstruction (LVOTO), severe mitral regurgitation, brady-arrhythmias, ventricular arrhythmias, or cardiac arrest [21,22].

Symptoms in the peripartum period do not differ from the general population. Younger women suffer more frequently from dyspnea. Despite the younger age, an appreciable percentage underwent invasive ventilation [23]. A retrospective study revealed a higher rate of TTS in pregnancy accompanied by an increased frequency of severe complications such as shock and respiratory insufficiency, requiring mechanical support [24]. The main issue is that TTS is confused with PPCM.

## 6. Definition and Diagnostic Criteria

Often TTS, especially its atypical forms, can be initially misdiagnosed and confused with acute coronary syndromes or PPCM after delivery. For this reason, over the years many diagnostic criteria have been proposed. The Mayo Clinic Criteria, despite being the most widely known, had gaps. The Takotsubo International Registry developed new international criteria—InterTAK. The most significant changes are that coronary artery disease can co-exist and that pheochromocytoma is not excluded from these criteria [25] (Table 2).

## 7. Types of TTS

The most common TTS type and widely recognized form is the (a) apical ballooning also known as the typical TTS form, which occurs in the majority of cases (81.7%). Over the past years, atypical TTS types have been increasingly recognized. These include the (b) midventricular (14.6%), (c) basal (2.2%), and (d) focal wall motion patterns (1.5%) [26]. The apex has the highest density of b-receptors and the lowest sympathetic innervations. This might explain why the apex is more susceptible to high levels of catecholamines provoking additional negative inotropic effects [27].

Recently, subjects with atypical presentation of TTS have different phenotypes. They are younger, have neurological comorbidities, lower brain natriuretic peptide (BNP) values, and less impaired ejection fraction accompanied more often with ST depression. Young pregnant women or postpartum with TTS can be classified in this form [26]. Biventricular TTS is reported, such as isolated right ventricle and global variant [28,29,30]. Involvement of the right ventricle predicts a worse outcome and occurs in about 30% of TTS [31]. The four types of TTS in ventriculography are illustrated in Figure 6 [4].

## 8. Triggers

Undoubtedly, TTS is linked with a stressful event. When the syndrome was first described, the trigger involved emotional trauma. Later, physical stressors have been noted and finally are more common than emotional ones. Notably, the TTS trigger in males is often physical, while in females it is emotional [4]. Emotional stressors include grief (death), severe conflicts (divorce, estrangement), fear, anger, anxiety, financial/employment issues, embarrassment, natural disasters, etc. Interestingly TTS occurred with positive triggers such as a surprise party and jackpot winning [32,33,34,35]. Physical triggers are associated with medical problems and procedures, pregnancy, postpartum, cesarean section, critical illness, exogenous drug administration, near drowning, alcohol, extreme activities, and several nervous system conditions [36,37,38,39,40,41,42]. On the other hand, one-third of cases had no identifiable stress to justify the occurrence [43].

## 9. Risk Factors

### 9.1. Hormonal

The high prevalence of TTS in postmenopausal females strongly suggests a hormonal effect. Females older than 55 years old have five-fold risk of suffering from TTS when compared with those younger than 55 years [44]. Estrogens can attenuate catecholamine-mediated vasoconstriction and decrease the sympathetic response to mental stress in perimenopausal women [45,46]. They also improve coronary blood flow in the microcirculation [47]. A recent study reveals the cardioprotective role of estrogen, supporting that normal depletion after delivery can lead to TTS in the peripartum period [16].

### 9.2. Genetic

Observational studies in family members who suffered from TTS suggest that genetic predisposition may exist. Many genetic pathways have been studied and proposed associated with β_1_, β_2_, and α_2_ receptor variants and polymorphism [48].

### 9.3. Psychiatric and Neurological Disorders

Patients presenting with TTS have a high prevalence of psychiatric and neurological disorders.

A percentage of 42% had a psychiatric diagnosis, mainly depression [4]. Interestingly, TTS patients were found to have type D personality which is characterized by negative emotions and higher cardiovascular risk [49]. Patients suffering from depression have excessive norepinephrine response to stress and reduced reuptake of norepinephrine [50].

On the other side, acute or chronic neurological history was reported in 27% of patients, especially stroke and subarachnoid hemorrhage [4,51,52].

## 10. Diagnostic Workout

Patients presenting with STEMI must proceed urgently for coronary angiography and ventriculography to exclude myocardial infarction. When presenting with non-STEMI the InterTAK score must be applied (Table 3). A score ≤70 suggests a low to intermediate probability of TTS, while a score of ≥70 is more likely to suffer from TTS. Patients with low probability should undergo angiography and those with high probability must initially have a transthoracic echocardiography. A diagnostic algorithm to guide clinicians has been proposed and is shown in Figure 7 [53].

Most patients are presented with abnormal ECG findings. All types of ST abnormalities can be seen. The most common finding is ST elevation (44%) followed by T wave inversion (41%), ST depression (8%), and LBBB (5%) [4].

In young women during pregnancy and the postpartum period, it is crucial to differentiate TTS from PPCM.

In summary, some features to distinguish these entities are shown in Table 4 [23].

### 10.1. Biomarkers

All patients suffering from TTS have elevated levels of troponin. Compared to acute coronary syndromes (ACS), the peaks of troponin and creatine kinase are significantly lower. High troponin predicts a worse outcome [4]. Remarkably, myocardial necrosis biomarkers do not reflect the large area of the affected LV regions in TTS patients. In contrast, BNP and Prohormone of brain natriuretic peptide (ProBNP) are substantially increased in TTS [54].

### 10.2. Echocardiography

Echocardiography is the cornerstone imaging modality to evaluate patients with suspected TTS. The availability is very high and can be performed bedside. Furthermore, due to no radiation exposure, it is safe for all patients including pregnant women. The identification of all variants of TTS can be made with careful observation and suspicion. The variants are described above. The regional wall motion abnormalities usually do not correspond to a single coronary artery territory. The right ventricle can also be involved and is presented with dilatation and hypokinesia or akinesia of the free wall [55,56]. The ejection fraction is an independent prognostic marker which is assessed accurately by echocardiogram. Dynamic LVOTO and systolic anterior motion of the mitral leaflet caused by hyperkinetic basal segments are essential to be diagnosed in patients with cardiogenic shock [57]. Severe mitral regurgitation is present in 14–25% of TTS patients due to dysfunction or displacement of the papillary muscles and tethering of the mitral valve [58]. Thrombus formation and covered rupture of LV free wall can be identified with echocardiography. In peripartum women, an echocardiogram is also used to differentiate TTS from PPCM.

### 10.3. Coronary Angiography

Coronary angiography and ventriculography are required in the majority of cases to differentiate TTS from acute coronary syndromes and especially from STEMI. The diagnostic algorithm is presented in Figure 3. When coronary artery disease and TTS coexist, careful assessment of ventriculography and angiography may distinguish the two pathologies. Approximately 20% of patients suffering from TTS have LVOTO. A hemodynamic assessment of the pressure gradient is required [59]. During pregnancy, radiation exposure is harmful to the fetus, especially in the early stage. For this reason, careful evaluation of the severity and necessity of the exam must be held.

### 10.4. Cardiac Computed Tomography Angiography

An alternative imaging modality instead of coronary angiography in specific circumstances is cardiac computed tomography angiography (CCTA). This exam is useful for patients with life-threatening conditions where classic angiography may pose a high risk of complications. It can also be performed in stable patients with a high probability of TTS [60]. CCTA provides reliable information about coronary artery anatomy and LV contraction. Due to radiation and the contrast agent, it is not preferable for pregnant women but can be used in the postpartum period.

### 10.5. Cardiac Magnetic Resonance

Cardiac magnetic resonance (CMR) imaging is a novel modality in TTS. It is difficult to use in the acute phase. In addition, the availability of CMR is not very high. Nevertheless, CMR is useful in the subacute stage. In dysfunctional LV regions, the absence of late gadolinium enhancement is crucial to distinguish TTS from other pathologic situations, such as ACS or myocarditis. CMR compared with echocardiography is more sensitive to reveal right ventricle involvement [31]. Gadolinium CMR should be used only if it is going to significantly improve diagnosis, and fetal or maternal outcome [61].

## 11. Complications and Prognosis

For many years, TTS was considered a benign disease with full recovery. Contemporary studies reveal controversial results. Cardiogenic shock and death rates are similar to ACS patients [62,63]. Despite the reversibility of the syndrome, electrical instability and hemodynamic deterioration increase the risk of severe in-hospital complications. Serious complications include death, heart failure, cardiogenic shock, LVOTO, atrioventricular block, and lethal ventricular arrhythmias [4]. In summary, the in-hospital complications are described in Table 5.

Evidence from the largest cohort to date suggests that the estimation of death is 5.6% per patient-year and in-hospital mortality is 1–4.5% [4]. Patients who suffer from TTS may have a new event in about 5% of cases [32].

## 12. Management

Patients presenting with mild TTS, without heart failure, are treated generally with Angiotensin-converting enzyme inhibitors (ACEi) and b-blockers. Note that ACEi and Angiotensin receptor blockers (ARBs) are contraindicated in pregnancy but not in the postpartum period. Inotropes such as adrenaline, noradrenaline, dobutamine, and milrinone must be avoided. In cases involving heart failure and pulmonary edema, the administration of diuretics and nitroglycerin (in the absence of LVOTO) should be considered in addition to previous therapy. Severely ill patients with hypotension-cardiogenic shock are at high risk. When catecholamines are necessary, the mortality rises up to 20% [64]. These patients require intensive care unit monitoring and therapy. The primary consideration is to distinguish whether the shock is due to pump failure or LVOTO. When the cause is pump failure, the option is Levosimendan and mechanical support with a left ventricular assist device (LVAD) or extracorporeal membrane oxygenation [65]. In contrast, if LVOTO is the primary disorder treatment should be cautious fluid administration, short-acting b-blockers, and LVAD. Diuretics, nitroglycerin, and intra-aortic balloon pumps should be avoided. There is no customized therapy for each type of TTS. Reverse TTS can rarely be complicated with LVOTO and if shock is presented, pump failure probably is the main cause.

Thrombus formation or embolism has a high incidence in patients suffering from TTS. Due to hypercoagulability, which is common in pregnancy, patients with low ejection fraction and apex involvement have a 3.3% probability of thrombus formation. In this condition, anticoagulants are required [66].

Complications are treated according to general guidelines. Long-term therapy includes ACE inhibitors or ARB, hormone replacement wherever required in postmenopausal women, and combined psycho-cardiac rehabilitation when psychiatric disorders exist.

## 13. Conclusions

In conclusion, TTS is a relatively new disease and the acknowledgment is important to provide proper therapy. Due to a better understanding of the syndrome cardiologists must have high clinical suspicion to diagnose TTS. The main difference between the general population and peripartum women is that the latter seem to be more susceptible to suffer from reverse TTS. Most of the young females present with dyspnea and not angina. PPCM must be excluded, due to similar symptoms. The ejection fraction of our patients who were diagnosed with reverse TTS was less impaired compared with the apical type. This is in accordance with the current literature. Additionally, in reverse TTS the patient is unlikely to exhibit LVOTO due to basal hypo/akinesia. Another difference from current studies is that our patients did not have neurological disorders.

We should mention that one-third of our cases were initially misdiagnosed. Our opinion is that many cases presenting with the atypical forms of TTS may not be recognized appropriately. The basal type of TTS as reported above is rare, about 2.2% of all syndromes. While the incidence of TTS in peripartum is still unknown, in one study it is 0.003%. After our study, we suggest that reverse TTS is not so rare, especially in this group of patients, due to the inability to identify this type. However, data must be derived from randomized trials and studies to clarify the incidence and guide the management in order to have better outcomes.

## Figures and Tables

**Figure 1 healthcare-12-01602-f001:**
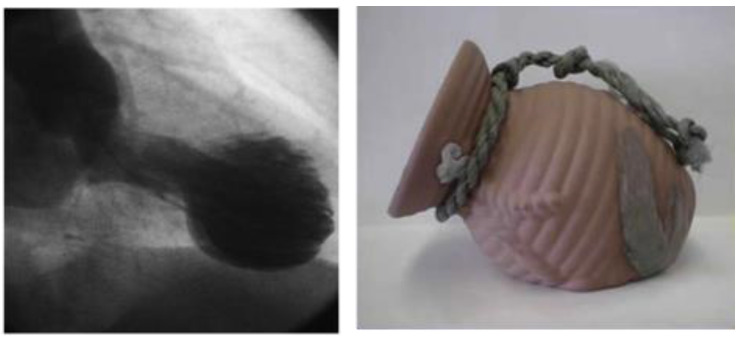
The Japanese octopus trap (right). Adapted from Santos I.

**Figure 2 healthcare-12-01602-f002:**
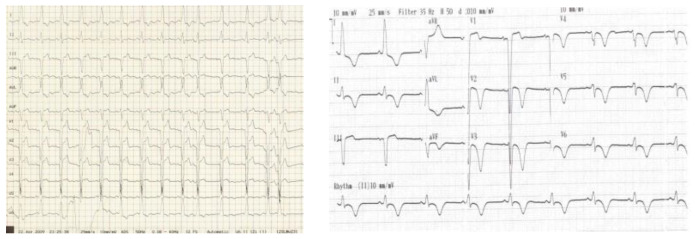
Abnormal ECG in peripartum women with TTS.

**Figure 3 healthcare-12-01602-f003:**
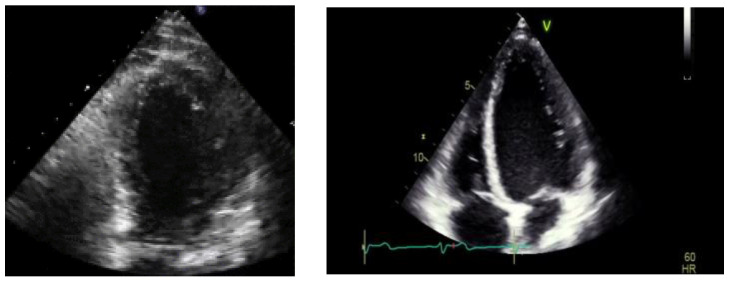
Four chambers in TTS patient at systole.

**Figure 4 healthcare-12-01602-f004:**
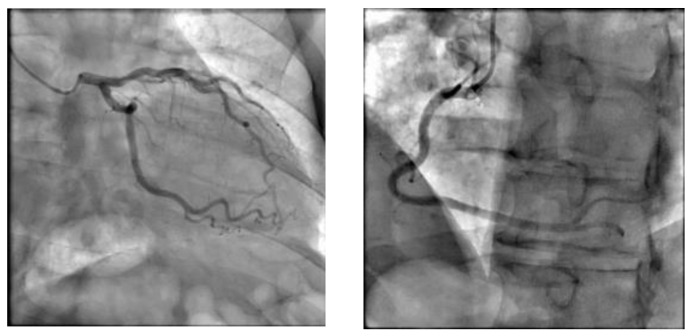
Normal coronary angiography in patient with TTS.

**Figure 5 healthcare-12-01602-f005:**
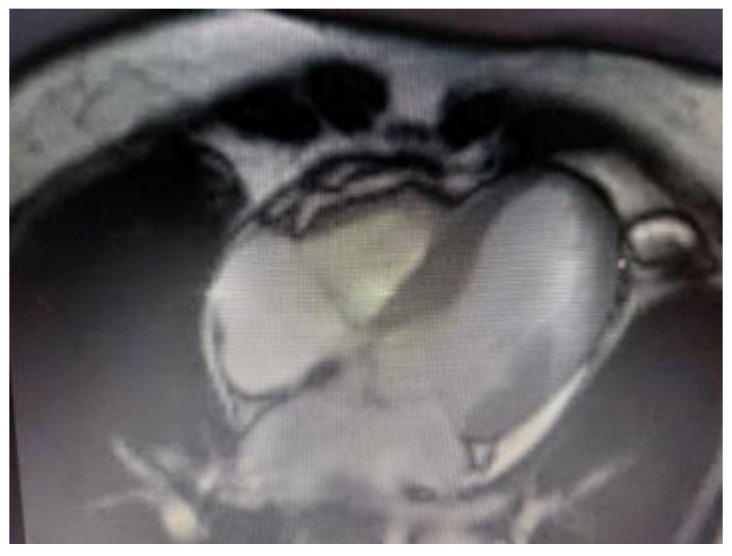
MRI image in apical balloon TTS during systole.

**Figure 6 healthcare-12-01602-f006:**
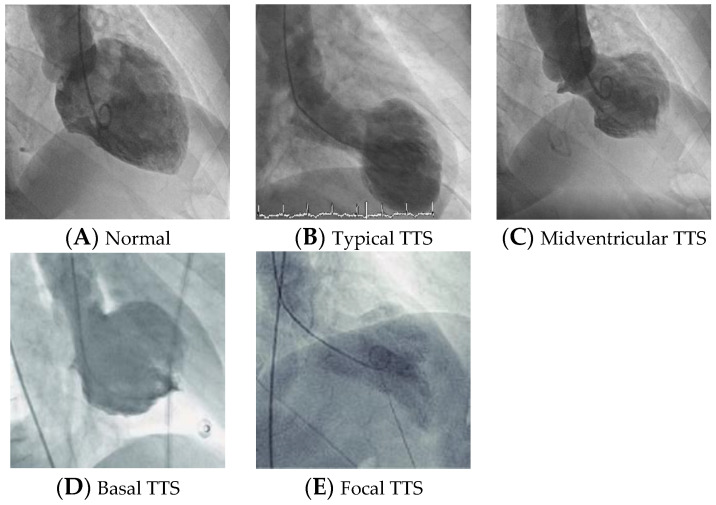
The four different types of Takotsubo syndrome. (**A**) Normal ventriculography. (**B**) Apical TTS. (**C**) Midventricular TTS. (**D**) Basal–Reverse TTS. (**E**) Focal TTS.

**Figure 7 healthcare-12-01602-f007:**
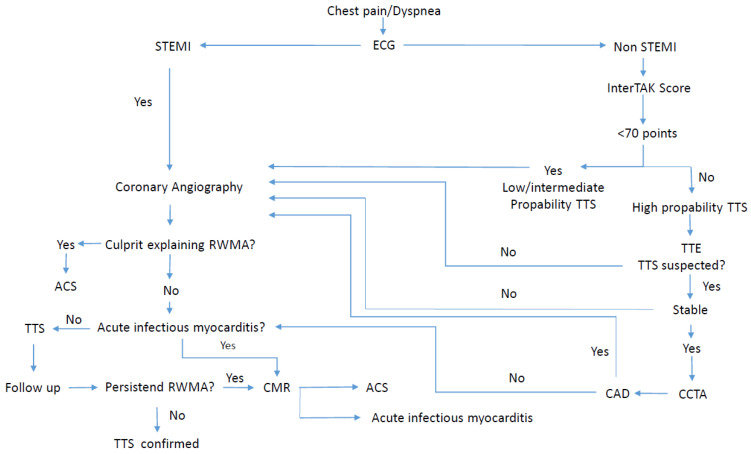
Diagnostic algorithm of Takotsubo syndrome. ACS, acute coronary syndrome; CAD, coronary artery disease; CCTA, coronary computed tomography angiography; CMR, cardiac magnetic resonance; ECG, electrocardiogram; InterTAK, International Takotsubo Registry; RWMA, regional wall motion abnormality; TTE, transthoracic echocardiography; TTS, Takotsubo syndrome.

**Table 1 healthcare-12-01602-t001:** Patients features.

Age	Symptoms	EF (%)	TTS Type	CA/CCTA	Complications	ICU Days	ProBNP/Troponin	Outcome	Follow Up
28	Dyspnea	46	Reverse	CCTA	No	3	Both positive	Good	Alive w/o complications
35	Dyspnea	40	Apical	Refused	No	4	Both positive	Good	Alive w/o complications
38	Chest pain	44	Reverse	CA	No	3	Both positive	Good	Alive w/o complications
40	Dyspnea	38	Apical	Refused	No	4	Both positive	Good	Lost to follow up
40	Dyspnea	43	Reverse	CA	No	3	Both positive	Good	Alive w/o complications
33	Chest pain	33	Apical	CA	No	4	Both positive	Good	Alive w/o complications
37	Dyspnea	35	Apical	Refused	No	3	Both positive	Good	Lost to follow up
35	Dyspnea	38	Apical	CCTA	No	4	Both positive	Good	Alive w/o complications
37	Dyspnea	34	Apical	CA	No	4	Both positive	Good	Alive w/o complications

EF: Ejection fraction (calculated by Simpson Method), TTS: Takotsubo syndrome, CA: coronary angiography, CCTA: cardiac computed tomography. angiography, ICU: Intensive Care Unit, w/o: without.

**Table 2 healthcare-12-01602-t002:** InterTAK diagnostic criteria.

International Takotsubo Diagnostic Criteria (InterTAK Diagnostic Criteria)
1. Patients show transient ^a^ left ventricular dysfunction (hypokinesia, akinesia, or dyskinesia) presenting as apical ballooning or midventricular, basal, or focal wall motion abnormalities. Right ventricular involvement can be present. Besides these regional wall motion patterns, transitions between all types can exist. The regional wall motion abnormality usually extends beyond a single epicardial vascular distribution. However, rare cases can exist where the regional wall motion abnormality is present in the subtended myocardial territory of a single coronary artery (focal TTS). ^b^
2. An emotional, physical, or combined trigger can precede the TTS event, but this is not obligatory.
3. Neurologic disorders (eg, subarachnoid hemorrhage, stroke/transient ischaemic attack, or seizures), as well as pheochromocytoma, may serve as triggers for TTS.
4. New ECG abnormalities are present (ST-segment elevation, ST-segment depression, T-wave inversion, and QTc prolongation); however, rare cases exist without any ECG changes.
5. Levels of cardiac biomarkers (troponin and creatine kinase) are moderately elevated in most cases; significant elevation of brain natriuretic peptide is common.
6. Significant coronary artery disease is not a contradiction in TTS.
7. Patients have no evidence of infectious myocarditis. ^b^
8. Postmenopausal women are predominantly affected.

^a^ Wall motion abnormalities may remain for a prolonged period of time or documentation of recovery may not be possible. For example, death before evidence of recovery is captured. ^b^ Cardiac magnetic resonance imaging is recommended to exclude infectious myocarditis and diagnosis confirmation of Takotsubo syndrome. Adapted from Ghadri [3].

**Table 3 healthcare-12-01602-t003:** InterTAK score.

InterTAK Diagnostic Score	
Female sex	25 points
Emotional stress	24 points
Physical stress	13 points
No ST-Segment depression	12 points
Psychiatric disorders	11 points
Neurologic disorders	9 points
QT_C_ prolongation	6 points

Adapted from Ghadri [3].

**Table 4 healthcare-12-01602-t004:** Differentiation of TTS and PPCM.

Differentiation Features between TTS and PPCM		
Criteria	TTS	PPCM
Onset	Commonly after C-section. Also describe in antepartum and late postpartum period	Heart failure in last month of pregnancy/within 5 months postpartum
Risk factor	More (twin gestation), multiple pregnancy	Less
Clinical features	Mimics MI, pericarditis, Heart failure, PTE	Mostly presents as heart failure
ECG	New onset ST segment changes or T wave inversion.	No pattern of ST segment changes
Cardiac enzymes	Higher elevation of CKMB, Troponin-T	Lesser elevations of CKMB, Troponin-T
2D ECHO	Apical hypokinesia (Typical TTCM), Mid ventricular, basal hypokinesia (Reverse TTS)	Global hypokinesia
Prognosis	More favourable	Less favourable
Recovery	Generally, within 1 month of delivery	Delayed recovery

TTS: Takotsubo Syndrome, PPCM: Peripartum cardiomyopathy, MI: Myocardial infarction, PTE: Pulmonary Thromboembolism. Adapted from Garg, et al. [23]

**Table 5 healthcare-12-01602-t005:** In-hospital complications.

In-Hospital Complications		
Frequent	Moderate	Rare
Acute heart failure (12–45%) LVOTO (10–25%) Mitral regurgitation (14–25%) Cardiogenic shock (6–20%)	Atrial fibrillation (5–15%) LV thrombus (2–8%) Cardiac arrest (4–6%) AV block (5%)	Tachyarrhythmia (2–5%) Bradyarrhythmia (2–5%) Torsade de pointes (2–5%) Death (1–4.5%) VT/VF (3%) Ventricular septal defect (<1%)

AV: atrioventricular block, LV: left ventricle, LVOTO: left ventricular outflow tract obstruction, VT: ventricular tachycardia, VF: ventricular fibrillation. Adapted from Templin C et al. [64]

## Data Availability

Data are contained within the article.

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
