# Peer review of "Takotsubo Syndrome: Differences between Peripartum Period and General Population"

_healthcare, 2024, doi:10.3390/healthcare12161602_

Round 1

Reviewer 1 Report

Comments and Suggestions for Authors

Tzerefos et al. in their review article described a rare disease known as Takotsubo cardiomyopathy. 

Here’s some of my comments

1.             Title:  The title did not reflect what was discussed in length in the manuscript. While I am expecting a vast discussion on reverse Takotsubo cardiomyopathy in pregnancy, but the focus is on general Takotsubo cardiomyopathy not restricted to that of pregnancy. Suggest to change either the title or modify the review accordingly.

2.             Also, reverse Takotsubo cardiomyopathy, which is a rare variant of TTS, was not discussed in detail as reflected in the Title. Suggest to be more focus on this variant if this should be the scope of this review.

3.             Line 49 – 50. The aim of this review is not very clear. Which part of the review differs significantly from that was previously published?

4.             Figure 2 and Figure 3. They are exactly the same as that published in other journals. Have the authors acquired permission from the respective publishers?

5.             Suggest to include a Table including all cases of TTS in pregnancy (including all the 9 cases from the authors’ hospital) were published in the previous years.

6.             Line 387, since CCTA is contradicted in pregnant women, why was it offered to two of the women suspected TTS?

7.             I am wondering how the conclusion “this combination is not rare, but underestimated” was drawn by the authors? On what clinical grounds?

Comments on the Quality of English Language

~

Reviewer 2 Report

Comments and Suggestions for Authors

This article provides a timely review of a rare pregnancy complication, takotsubo cardiomyopathy. The following are suggested to improve this paper:

1. Present you own cases first before going into literature review. Start with clinical features, clinical presentation, and diagnostic work up followed by follow up information (are they alive without complications, alive with complications, died of disease, or died or other cause, or lost to follow up).

2. Please show representative images of diagnostic workup (ECG, MRI) from your patients.

3. Please include the discussion of reverse of reverse takotsubo in the introduction.

4. Please be consistent with the use of takotsubo cardiomyopathy. There are areas where it is misspelled as Tako Tsubo.

5. Please improve the tables and remove the grid lines within the table and align left.

6. Please provide references for the contents of the tables and images when taken from a published source.

7. Please correct some major grammar and spelling errors.

Comments on the Quality of English Language

The English language requires moderate to extensive editing.

Reviewer 3 Report

Comments and Suggestions for Authors

Chapter "Our experience" - you need to improve and to extend this chapter. Please insert ideas regarding: identified risk factors, biochemical abnormalities associated, echocardiographic findings. Considering the small number of cases presented, it would be important to expand aspects regarding the clinical presentation and other data which helped you determine the final diagnosis, as well as differential diagnoses you have considered.

Reviewer 4 Report

Comments and Suggestions for Authors

The content of the article is not aligned with the title and purpose of the research.

The aim of the study is to discuss whether Takotsubo Syndrome is a rare occurrence during pregnancy. The epidemiology is presented briefly and does not provide information on the prevalence of the condition. Given the increasing interest in this issue worldwide, research is ongoing in various countries and their findings are published in PubMed. It would be beneficial for the authors to conduct a more comprehensive literature review and include these data in their article. The sections on pathophysiology, symptoms, definition, criteria, types, risk factors, and triggers are described in great detail, which may not be relevant to the focus of the study or the title. These sections, along with the figures in the paper, are similar to those published by other researchers with minor variations (Ghadri et al., 2020).International expert consensus document on Takotsubo cardiomyopathy (part I): clinical characteristics, diagnostic criteria, and pathophysiology (Eur Heart J. 2018;49(39):2032-46 ). This may be considered plagiarism. 

Own data is presented sparsely, the topic is not disclosed. You should add the criteria by which the diagnosis was made, provide data from echocardiography and other studies that confirm the diagnosis of Takotsubo syndrome in your patients.

Round 2

Reviewer 2 Report

Comments and Suggestions for Authors

This revised version of the manuscript has greatly improved the readability and presentation of the authros' study. Please address the minor comments below prior to publication:

1. Please mention if your study has an IRB approval and include the IRB number and date.

2. Please discuss if there is any difference in management among the different types of takotsubo.

3. Please avoid very long and passive sentences, and correct some minor grammatical errors.

Comments on the Quality of English Language

Good quality of English language use. However, some minor grammatical and spelling errors are still present.

Reviewer 4 Report

Comments and Suggestions for Authors

The title states: "Takotsubo syndrome: differences between the periportal period and general population". The differences should be reflected in the conclusions and conclusion.
